# Learning to Make Decisions via Submodular Regularization

**Ayya Alieva**
Stanford University
ayya@stanford.edu

**Aiden Aceves**
Caltech
aaceves@caltech.edu

**Jialin Song**
Caltech
jssong@caltech.edu

**Stephen Mayo**
Caltech
steve@caltech.edu

**Yisong Yue**
Caltech
yyue@caltech.edu

**Yuxin Chen**
University of Chicago
chenyuxin@uchicago.edu

## Abstract

Many sequential decision making tasks can be viewed as combinatorial optimization problems over a large number of actions. When the cost of evaluating an action is high, even a greedy algorithm, which iteratively picks the best action given the history, is prohibitive to run. In this paper, we aim to learn a greedy heuristic for sequentially selecting actions as a surrogate for invoking the expensive oracle when evaluating an action. In particular, we focus on a class of combinatorial problems that can be solved via submodular maximization (either directly on the objective function or via submodular surrogates). We introduce a data-driven optimization framework based on the *submodular-norm* loss, a novel loss function that encourages the resulting objective to exhibit *diminishing returns*. Our framework outputs a surrogate objective that is efficient to train, approximately submodular, and can be made permutation-invariant. The latter two properties allow us to prove strong approximation guarantees for the learned greedy heuristic. Furthermore, our model is easily integrated with modern deep imitation learning pipelines for sequential prediction tasks. We demonstrate the performance of our algorithm on a variety of batched and sequential optimization tasks, including set cover, active learning, and data-driven protein engineering.

## 1 Introduction

In real-world automated decision making tasks we seek the optimal set of actions that jointly achieve the maximal utility. Many of such tasks — either deterministic/non-adaptive or stochastic/adaptive — can be viewed as combinatorial optimization problems over a large number of actions. As an example, consider the active learning problem where a learner seeks the maximally-informative set of training examples for learning a classifier. The utility of a training set could be measured by the mutual information (Lindley, 1956) between the training set and the remaining (unlabeled) data points, or by the expected reduction in generation error if the model is trained on the candidate training set. Similar problems arise in a number of other domains, such as experimental design (Chaloner and Verdinelli, 1995), document summarization (Lin and Bilmes, 2012), recommender system (Javdani et al., 2014), and policy making (Runge et al., 2011).

Identifying the optimal set of actions (e.g., optimal training sets, most informative experiments) amounts to evaluating the expected utility over a combinatorial number of candidate sets. When the underlying model class is complex and the evaluation of the utility function is expensive, these tasks are notoriously difficult to optimize (Krause and Guestrin, 2009). For a broad class of decision making problems whose optimization criterion is to maximize the decision-theoretic *value of information* (e.g., active learning and experimental design), it has been shown that it is possible to design surrogate objective functions that are (approximately) submodular while being aligned with the original objective at the optimal solutions (Javdani et al., 2014; Chen et al., 2015b; Choudhury et al., 2017). Here, the information gathering policies no longer aim to directly optimize the target objective value, but rather choose to follow a greedy trajectory governed by the surrogate function

that is much cheaper to evaluate. These insights have led to principled algorithms that enable significant gains in the efficiency of the decision making process, while enjoying strong performance guarantees that are competitive with the optimal policy.

Despite the promising performance, a caveat for these "submodular surrogate"-based approaches is that it is often challenging to engineer such a surrogate objective without an ad-hoc design and analysis that requires trial-and-error (Chen et al., 2015b; Satsangi et al., 2018). Furthermore, for certain classes of surrogate functions, it is NP-hard to compute/evaluate the function value (Javdani et al., 2014). In such cases, even a greedy policy, which iteratively picks the best action given the (observed) history, can be prohibitively costly to design or run. Addressing this limitation requires more automated or systematic ways of designing (efficient) surrogate objective functions for decision making.

**Overview of main results.** Inspired by contemporary work in data-driven decision making, we aim to learn a greedy heuristic for sequentially selecting actions. This heuristic acts as a surrogate for invoking the expensive oracle when evaluating an action. Our key insight is that many practical algorithms can be interpreted as greedy approaches that follow an (approximate) submodular surrogate objective. In particular, we focus on the class of combinatorial problems that can be solved via submodular maximization (either directly on the objective function or via a submodular surrogate). We highlight some of the key results below:

- Focusing on utility-based greedy policies, we introduce a data-driven optimization framework based on the "*submodular-norm*" loss, which is a novel loss function that encourages learning functions that exhibit "diminishing returns". Our framework, called LEASURE (Learning with Submodular Regularization), outputs a surrogate objective that is efficient to train, approximately submodular, and can be made permutation-invariant. The latter two properties allow us to prove approximation guarantees for the resulting greedy heuristic.

- We show that our approach can be easily integrated with modern imitation learning pipelines for sequential prediction tasks. We provide a rigorous analysis of the proposed algorithm and prove strong performance guarantees for the learned objective.

- We demonstrate the performance of our approach on a variety of decision making tasks, including set cover, active learning for classification, and data-driven protein design. Our results suggest that, compared to standard learning-based baselines: (a) at training time, LEASURE requires significantly fewer oracle calls to learn the target objective (i.e., to minimize the approximation error against the oracle objective); and (b) at test time, LEASURE achieves superior performance on the corresponding optimization task (i.e., to minimize the regret for the original combinatorial optimization task). In particular, LEASURE has shown promising performance in the protein design task and will be incorporated into a real-world protein design workflow.

## 2 RELATED WORK

**Near-optimal decision making via submodular optimization.** Submodularity is a property of a set function that has a strong relationship with diminishing returns, and the use of submodularity has wide applications from information gathering to document summarization (Leskovec et al., 2007; Krause et al., 2008; Lin and Bilmes, 2011; Krause and Golovin, 2014). The maximization of a submodular function has been an active area of study in various settings such as centralized (Nemhauser et al., 1978; Buchbinder et al., 2014; Mitrovic et al., 2017), streaming (Badanidiyuru et al., 2014; Kazemi et al., 2019; Feldman et al., 2020), continuous (Bian et al., 2017b; Bach, 2019) and approximate (Horel and Singer, 2016; Bian et al., 2017a). Variants of the greedy algorithm, which iteratively selects an element that maximizes the marginal gain, feature prominently in the algorithm design process. For example, in the case of maximizing a monotone submodular function subject to a cardinality constraint, it is shown that the greedy algorithm achieves an approximation ratio of $(1 - 1/e)$ of the optimal solution (Nemhauser et al., 1978).

In applications where we need to make a sequence of decisions, such as information gathering, we usually need to adapt our future decisions based on past outcomes. Adaptive submodularity is the corresponding property where an adaptive greedy algorithm enjoys a similar guarantee for maximizing an adaptive submodular function (Golovin and Krause, 2011). Recent works have explored optimizing the value of information (Chen et al., 2015b) and Bayesian active learning (Javdani et al., 2014; Chen et al., 2017a) with this property. Another line of related work is online setting (typically

bandits), which is grounded in minimizing cumulative regret (Radlinski et al., 2008; Streeter et al., 2009; Yue and Guestrin, 2011; Ross et al., 2013; Yu et al., 2016; Hiranandani et al., 2020).

**Learning submodular functions.** Early work focused on learning non-negative linear combinations of submodular basis functions (Yue and Joachims, 2008; El-Arini et al., 2009; Yue and Guestrin, 2011; Sipos et al., 2012), which was later generalized to mixtures of "submodular shells" (Lin and Bilmes, 2012). Deep submodular functions (Dolhansky and Bilmes, 2016) extend these ideas to more expressive compositional function classes by using sums of concave composed with modular functions. The theoretical question of the learnability of general submodular functions is analyzed in Balcan and Harvey (2018). Our goal is to encourage submodularity via regularization, rather than via hard constraints on the function class design.

**Learning to optimize via imitation learning.** Rather than first learning a submodular function and then optimizing it, one can instead learn to directly make decisions (e.g., imitate the oracle greedy algorithm). This area builds upon imitation learning, which learns a policy (i.e., a mapping from states to actions) *directly* from examples provided by an expert (e.g., an expensive computational oracle, or a human instructor) (Chernova and Thomaz, 2014). Classic work on imitation learning (e.g., the Dataset Aggregation (DAgger) algorithm (Ross et al., 2011)) reduce the policy learning problem to the supervised learning setting, which has been extended to submodular optimization by imitating the greedy oracle method (Ross et al., 2013). More generally, learning to optimize has been applied generically to improve combinatorial optimization solvers for focused distributions of optimization problems (He et al., 2014; Song et al., 2018; Khalil et al., 2016; Balunovic et al., 2018; Gasse et al., 2019; Song et al., 2020). Our approach bridges learning to optimize and learning submodular functions, with a focus on learning surrogate utilities using submodular regularization.

**Learning active learning.** Our approach is applicable to active learning, and so is related to work on learning active learning. The closest line of work learns a utility function as a surrogate for improvement in classifier accuracy (Konyushkova et al., 2017; Liu et al., 2018), which is then used as the decision criterion. However, prior work either used restricted function classes (Konyushkova et al., 2017), or very expressive function classes that can be hard to fit well (Liu et al., 2018). Our work can be viewed as a direct extension of this design philosophy, where we aim to reliably learn over expressive function classes using submodular regularization. Other related work do not directly learn an active learning criterion, instead encouraging sample diversity using submodularity (Wei et al., 2015) or the gradient signal from the classifier (Ash et al., 2020).

## 3 BACKGROUND AND PROBLEM STATEMENT

### 3.1 DECISION MAKING VIA SUBMODULAR SURROGATES

Given a ground set of items $\mathcal{V}$ to pick from, let $u : 2^{\mathcal{V}} \to \mathbb{R}$ be a set function that measures the *value* of any given subset[1] $A \subseteq \mathcal{V}$. For example, for experimental design, $u(A)$ captures the utility of the output of the best experiment; for active learning $u(A)$ captures the generalization error after training with set $A$. We denote a policy $\pi : 2^{\mathcal{V}} \to \mathcal{V}$ to be a partial mapping from the set/sequence of items already selected, to the next item to be picked. We use $\Pi$ to denote our policy class. Each time a policy picks an item $e \in \mathcal{V}$, it incurs a unit cost. Given the ground set $\mathcal{V}$, the utility function $u$, and a budget $k$ for selecting items, we seek the optimal policy $\pi$ that achieves the maximal utility:

$$\pi^* \in \arg\max_{\pi \in \Pi} u(S_{\pi,k}). \tag{1}$$

$S_{\pi,k}$ is the sequence of items picked by $\pi$: $S_{\pi,i} = S_{\pi,i-1} \cup \{\pi(S_{\pi,i-1})\}$ for $i > 0$ and $S_{\pi,0} = \emptyset$.

As we have discussed in the previous sections, many sequential decision making problems can be characterized as constrained monotone submodular maximization problem. In those scenarios $u$ is:

- **Monotone**: For any $A \subseteq \mathcal{V}$ and $e \in \mathcal{V} \setminus A$, $u(A) \leq u(A \cup \{e\})$.

- **Submodular**: For any $A \subseteq B \subseteq \mathcal{V}$ and $e \in \mathcal{V} \setminus B$, $u(A \cup \{e\}) - u(A) \geq u(B \cup \{e\}) - u(B)$.

---

[1]For simplicity, we focus on deterministic set functions in this section. Note that many of our results can easily extent to the stochastic, by leveraging the theory of adaptive submodularity (Golovin and Krause, 2011)

In such cases, a mypopic algorithm following the greedy trajectory of $u$ admits a near-optimal policy. However, in many real-world applications, $u$ is not monotone submodular. Then one strategy is to design a surrogate function $f : 2^{\mathcal{V}} \to \mathbb{R}$ which is:

- Globally aligning with $u$: For instance, $f$ lies within a factor of $u$: $f(A) \in [c_1 \cdot u(A), c_2 \cdot u(A))]$ for some constants $c_1, c_2$ and any set $A \subseteq \mathcal{V}$; or within a small margin with $u$: $f(A) \in [u(A) - \epsilon, u(A) + \epsilon]$ for a fixed $\epsilon > 0$ and any set $A \subseteq \mathcal{V}$;
- Monotone submodular: Intuitively, a submodular surrogate function encourages selecting items that are beneficial in the long run, while ensuring that the decision maker does not miss out any actions that are "surprisingly good" by following a myopic policy (i.e., future gains for any item are diminishing). Examples that fall into this category include machine teaching (Singla et al., 2014), active learning (Chen et al., 2015a), etc.

We argue that in real-world decision making scenarios—as validated later in Section 6—the decision maker is following a surrogate objective that aligns with the above characterization. In the following context, we will assume that such surrogate function exists. Our goal is thus to learn from an *expert policy* that behaves greedily according to such surrogate functions.

## 3.2 LEARNING TO MAKE DECISIONS

We focus on the regime where the expert policy is expensive to evaluate. Let $g : 2^{\mathcal{V}} \times \mathcal{V} \to \mathbb{R}$ be the score function that quantifies the benefit of adding a new item to an existing subset of $\mathcal{V}$. For the expert policy and submodular surrogate $f$ discussed in Section 3.1, $\forall A \subseteq \mathcal{V}$ and $e \in \mathcal{V}$:

$$g^{\text{exp}}(A, e) = f(A \cup \{e\}) - f(A).$$

For example, in the active learning case, $g^{\text{exp}}(A, e)$ could be the expert acquisition function that ranks the importance of labelling each unlabelled point, given the currently labelled subset. In the set cover case, $g^{\text{exp}}(A, e)$ could be the function that gives the score to each vertex and determines the next best vertex to add to the cover set. Given a loss function $\ell$, our goal is to learn a score function $\hat{g}$ that incurs the minimal expected loss when evaluated against the expert policy: $\hat{g} = \arg\min_g \mathbb{E}_{A,e}[\ell(g(A, e), g^{\text{exp}}(A, e))]$. Subsequently, the utility by the learned policy is $u(S_{\hat{\pi},k})$, where for any given history $A \subseteq \mathcal{V}$, $\hat{\pi}(A) \in \arg\max_{e \in \mathcal{V}} \hat{g}(A, e)$.

## 4 LEARNING WITH SUBMODULAR REGULARIZATION

To capture our intuition that a greedy expert policy tends to choose the most useful items, we introduce LEASURE, a novel regularizer that encourages the learned score function (and hence surrogate objective) to be submodular. We describe the algorithm below.

Given the groundset $\mathcal{V}$, let $f : 2^{\mathcal{V}} \to \mathbb{R}$ be any approximately submodular surrogate such that $f(A)$ captures the "usefulness" of the set $A$. The goal of a trained policy is to learn a score function $g : 2^{\mathcal{V}} \times \mathcal{V} \to \mathbb{R}$ that mimics $g^{\text{exp}}(A, x) = f(A \cup \{x\}) - f(A)$, which is often prohibitively expensive to evaluate exactly. Then, given any such $g$, we can define a greedy policy $\pi(A) = \arg\max_{x \in \mathcal{V}} g(A, x)$. With LEASURE, we aim to learn such function $g$ that approximates $g^{\text{exp}}$ well while being inexpensive to evaluate at test time. Let $D_{real} = \{(\langle A, x \rangle, y^{\text{exp}} = g^{\text{exp}}(A, x))\}_m$ be the gathered tuple of expert scores for each set-element pair. If the set $2^{\mathcal{V}} \times \mathcal{V}$ was not too large, the LEASURE could be trained on the randomly collected tuples $D_{real}$. However, $2^{\mathcal{V}}$ tends to be too large to explore, and generating ground truth labels could be very expensive. To leverage that, for a subset of set-element pairs in $D_{real}$ we generate a set of random supersets to form an unsupervised synthetic dataset of tuples $D_{synth} = \{(\langle A, x \rangle, \langle A', x \rangle) | A \preceq A', \langle A, x \rangle \in D_{real}\}_n$ where $A'$ denote a randomly selected superset of $A$. Define:

$$\text{Loss}(g, g^{\text{exp}}) = \sum_{\langle A, x \rangle, y^{\text{exp}} \in D_{real}} (y^{\text{exp}} - g(A, x))^2 + \lambda \sum_{(\langle A, x \rangle, \langle A', x \rangle) \in D_{synth}} \sigma([g(A', x) - g(A, x)]),$$

where $\lambda > 0$ is the regularization parameter and $\sigma$ is the sigmoid function. Intuitively, such regularization term will force the learned function $g$ to be close to submodular, as it will lead to larger losses every time $g(A', x) > g(A, x)$. If we expect $f$ to be monotonic, we also introduce a second regularizer $\text{ReLu}(-g(A', x))$ which pushes the learned function to be positive. Combined, the loss

function becomes (used in Line 11 in Algorithm 1):

$$\text{Loss}(g, g^{\text{exp}}) = \sum_{\langle A, x \rangle, y^{\text{exp}} \in D_{real}} (y^{\text{exp}} - g(A, x))^2 + \lambda \sum_{(\langle A, x \rangle, \langle A', x \rangle) \in D_{synth}} \sigma([g(A', x) - g(A, x)])$$
$$+ \gamma \sum_{\langle A', x \rangle \in D_{synth}} \text{ReLu}(-g(A', x)),$$

where $\gamma$ is another regularization strength parameter. Such loss should push $g$ to explore a set of approximately submodular, approximately monotonic functions. Thus, if $f$ exhibits the submodular and monotonic behavior, $g$ trained on this loss function should achieve a good local minima.

We next note that since $2^{\mathcal{V}}$ is too large to explore, instead of sampling random tuples for $D_{real}$, we use modified DAgger. Then $g$ can learn not only from the expert selections of $\langle A, x \rangle$, but it can also see the labels of the tuples the expert would not have chosen.

---

**Algorithm 1** Learning to make decisions via Submodular Regularization (LEASURE)

---

1: **Input**: Ground set $\mathcal{V}$, expert score function $g^{\text{exp}}$,
2: regularization parameters $\lambda, \gamma$, DAgger constant $\beta$, the length of trajectories $T$.
3: initialize $D_{real} \leftarrow \emptyset$
4: initialize $g$ to any function.
5: **for** $i = 1$ to $N$ **do**
6:     Let $g_i = g^{\text{exp}}$ with probability $\beta$.
7:     Sample a batch of $T-$step trajectories using $\pi_i(A) = x_i = \text{argmax}_{x \in \mathcal{V}} g_i(A, x)$.
8:     Get dataset $D_i = \{\langle A_i, x_i \rangle, g^{\text{exp}}(A_i, x_i)\}$ of labeled tuples on actions taken by $\pi_i$.
9:     $D_{real} \leftarrow D_{real} \bigcup D_i$.
10:     Generate synthetic dataset $D_{synth}$ from $D_{real}$.
11:     Train $g_{i+1}$ on $D_{real}$ and $D_{synth}$ using the loss function above.
12: **Output**: $g_{N+1}$

---

Algorithm 1 above describes our approach. A trajectory in Line 7 is a sequence of iteratively chosen tuples, $(\langle \emptyset, x_1 \rangle, \langle \{x_1\}, x_2 \rangle, \langle \{x_1, x_2\}, x_3 \rangle ..., \langle \{x_1, ..., x_{T-1}\}, x_T \rangle)$, collected using a mixed policy $\pi_i$. In Line 8, expert feedback of selected actions is collected to form $D_i$. Note that in some settings, even collecting exact expert labels $g^{\text{exp}}$ at train time could be too expensive. In that case, $g^{\text{exp}}$ can be replaced with a less expensive, noisy approximate expert $g_\epsilon^{\text{exp}} \approx g^{\text{exp}}$. In fact, all three of our experiments use noisy experts in one form or another.

## 5 ANALYSIS

**Estimating the expert's policy.** We first consider the bound on the loss of the learned policy measured against the expert's policy. Since LEASURE can be viewed as a specialization of DAGGER (Ross et al., 2011) for learning a submodular function, it naturally inherits the performance guarantees from DAGGER, which show that the learned policy efficiently converges to the expert's policy. Concretely, the following result, which is adapted from the original DAgger analysis, shows that the learned policy is consistent with the expert policy and thus is a *no-regret* algorithm:

**Theorem 1** (Theorem 3.3, Ross et al. (2011))**.** *Denote the loss of $\hat{\pi}$ at history state $H$ as $l(H, \hat{\pi}) := \ell(g(H, \hat{\pi}(H)), g^{exp}(H, \pi^{exp}(H)))$. Let $d_{\hat{\pi}}$ be the average distribution of states if we follow $\hat{\pi}$ for a finite number of steps. Furthermore, let $D_i$ be a set of $m$ random trajectories sampled with $\pi_i$ at round $i \in \{1, \ldots, N\}$, and $\hat{\epsilon}_N = \min_\pi \frac{1}{N} \sum_{i=1}^N \mathbb{E}_{H_i \sim D_i}[l(H_i, \hat{\pi})]$ be the training loss of the best policy on the sampled trajectories. If $N$ is $\mathcal{O}\left(T^2 \log(1/\delta)\right)$ and $m$ is $\mathcal{O}(1)$ then with probability at least $1 - \delta$ there exists a $\hat{\pi}$ among the $N$ policies, with $\mathbb{E}_{H \sim d_{\hat{\pi}}}[l(H, \hat{\pi})] \leq \hat{\epsilon}_N + \mathcal{O}\left(\frac{1}{T}\right)$.*

**Approximating the optimal policy.** Note that the previous notion of regret corresponds to the average difference in score function between the learned policy and the expert policy. While this result shows that LEASURE is consistent with the expert, it does not directly address how well the learned policy performs in terms of the gained utility. We then provide a bound on the expected value of the learned policy, measured against the value of the optimal policy. For specific decision making tasks where the oracle follows an approximately submodular objective, our next result, which is proved in the appendix, shows that the learned policy behaves near-optimally.

**Theorem 2.** *Assume that the utility function* $u$ *is monotone submodular. Furthermore, assume the expert policy* $\pi^{exp}$ *follows a surrogate objective* $f$ *such that for all* $A \subseteq \mathcal{V}$, $|f(A) - u(A)| < \epsilon_E$ *where* $\epsilon_E > 0$. *Let* $\hat{\epsilon}_N = \min_\pi \frac{1}{N} \sum_{i=1}^N l(H_i, \hat{\pi})$ *be the training loss of the best policy on the sampled trajectories. If* $N$ *is* $\mathcal{O}\left(T^2 \log(1/\delta)\right)$ *then with probability at least* $1 - \delta$, *the expected utility achieved by running* $\hat{\pi}$ *for* $k$ *steps is*

$$\mathbb{E}[u(S_{\hat{\pi},k})] \geq (1 - 1/e)\mathbb{E}[u(S_{\pi^*,k})] - k(\epsilon_E + \Delta_{\max}\hat{\epsilon}_N) - O(1).$$

A closely related work in approximate policy learning is by Ross et al. (2013), which also builds upon DAGGER to tackle policy learning for submodular optimization, via directly imitating the greedy oracle decision rather than learning a surrogate utility. One key difference is that their approach can only yield guarantees against an artificial benchmark (a set or list of simpler policies that each independently selects an item to add to the action set), whereas our theoretical guarantees are with respect to the optimal policy in our class.

## 6 EXPERIMENTS

In this section, we demostrate the performance of LEASURE on three diverse sequential decision making tasks, namely set cover (SC), learning active learning (LAL) and protein engineering (PE).

**Baselines.** We compare our approach to the Deep Submodular Function (DSF (Dolhansky and Bilmes, 2016)) and Deep Batch Active Learning by Diverse, Uncertain Gradient Lower Bounds (BADGE (Ash et al., 2020)). The DSF approach learns a submodular surrogate function $f : 2^{\mathcal{V}} \to \mathbb{R}$ that produces a score for each set $A \subset \mathcal{V}$. The architecture of the DSF forces the function $f$ to be exactly submodular, as opposed to LEASURE, which is only encouraged to be submodular through a regularizer. However, the architecture and the training procedure of the DSF are quite restrictive, which does not allow the DSF to explore a large domain during training and restricts how expressive it can be compared to a standard neural network. Moreover, DSF are restricted to small $\mathcal{V}$, and the number of parameters increases with the cardinality of $\mathcal{V}$. That is not true for LEASURE, which number of parameters grows with the dimensionality of elements in $\mathcal{V}$. This makes DSF useful for small datasets, but makes it prohibitively expensive to use on larger problems. In fact, we could not compare LEASURE to DSF on LAL or PE tasks, as it was not feasible to train DSF on these sets. For LAL experiment, we also compare with a recent deep active learning approach (Ash et al., 2020). Finally, we want to add that LEASURE can be seamlessly integrated with any standard Machine Learning library, and since the architecture of the learned policy in LEASURE is not restrictive, any available optimization trick can be used to achieve better performance. In fact, existing 'imitation learning'-based approaches for LAL, such as Liu et al. (2018), can be viewed as special cases of LEASURE (i.e. without regularization). On the other hand, DSF cannot be as easily implemented, and the standard libraries are not optimized for the DSF architecture.

### 6.1 SET COVER

Before testing our approach on a real-world scenario, we showcase its performance on a simple submodular and monotonic maximization problem. Set cover is a classical example: given a set of elements $U = \{1, 2, ..., n\}$ (called the universe) and a collection of $m$ sets $S = \{s_1, .., s_m\}$ whose union equals the universe, the set cover problem is to identify the smallest sub-collection of $S$ whose union equals the universe. Formulated as a policy learning problem, the goal is to learn the score function $g : 2^S \times S \to \mathbb{R}$ such that for any $S_l \subset S, x \in S$,

$$g(S_l, x) \approx g^{\exp}(S_l, x) = |\cup_{s \in S_l} s \cup x| - |\cup_{s \in S_l} s|.$$

Given $g$, we can then define a policy $\pi : 2^s \to S$ as $\pi(S_l) = \arg\max_{x \in S} g(S_l, x)$. During training, tuples $\{(S_l, x), g^{\exp}\}$ are collected, and then $g$ is trained on this set. We trained four different policies: a function $g$ parametrized by a neural network with $MSE(g, g^{\exp})$ as the loss, a function $g$ with the same MSE loss and just a monotonicity regularizer, a function $g$ trained using both monotonicity and submodular regularizers (LEASURE), as well as the Deep Submodular Function baseline (Dolhansky and Bilmes, 2016). We use a modified Deepset architecture (Zaheer et al., 2017) for modeling the permutation-invariant score networks $g$ in both the SC and the LAL tasks, and provide the details in Appendix B. Our dataset is the subset of the Mushroom dataset (Lim, 2015), consisting of 1000 sets. Each set contains 23 mushroom species, and there are a total of 119

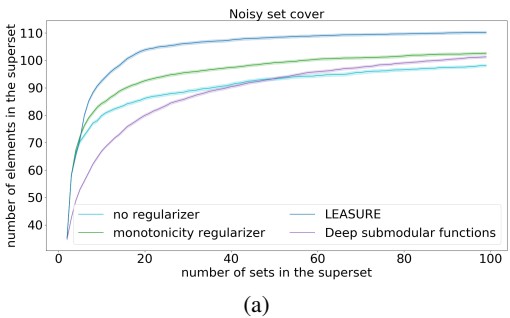 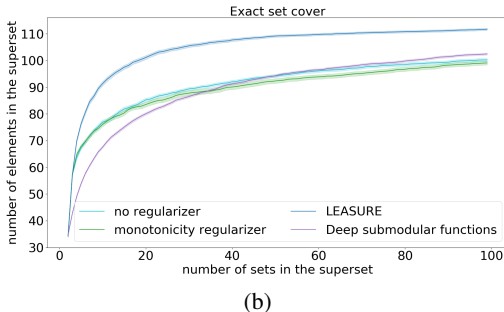

(a)                                                        (b)

Figure 1: Evaluating LEASURE against baselines on set cover instances

species. The goal is to train a policy to select the largest superset of these sets. We evaluate in two settings: Exact Set Cover, where we collect tuples $\{(S_l, x), g^{\exp}\}$ for training, and Noisy Set Cover, where we have access only to $\{(S_l, x), g_\epsilon^{\exp}\}$, where $g_\epsilon^{\exp}$ is a noisy score. The networks are trained on rollouts of length 20 (i.e. on sets $\{S_l : |S_l| \leq 20\}$), and tested on rollout of length up to 100.

Figure 1 show the value of set cover as a function of the size of the superset. LEASURE significantly outperforms other learned policies, although Deep Submodular Function generalizes better to larger rollout lengths – LEASURE gets most of its set cover gains in the first 10-20 selected points, while Deep Submodular Function continues to noticeably improve past the training rollout length. Note that in Figures 1a & 1b, the competing baselines all exhibit a "diminishing returns" effect, resulting in a concave-shaped value function. With a submodular-norm regularizer, LEASURE quickly identified the sets with large marginal gains. This observation aligns with our analysis in Section 5.

## 6.2 LEARNING ACTIVE LEARNING ON FASHION MNIST

In this section we demonstrate the performance of LEASURE on a real-world task that is not submodular or monotonic, but usually exhibits submodular and monotonic behaviour.

In active learning, there is a partially labelled dataset $S = \{S_l, S_u\}$, where $S_l$ is labelled and $S_u$ is unlabelled, and a policy $\pi : 2^S \to S$. The labelled subset $S_l$ can be used to infer from data (learn the image classifier, predict unlabelled protein fitness, etc). The goal of the policy is to select the smallest subset $S_\pi \subset S_u$ to label such that the accuracy of supervised learning from $S_\pi \cup S_l$ is maximized. Since selecting a subset is a prohibitively expensive combinatorial task, the policy is usually sequential. In particular, it selects points to add to $S_\pi$ one by one (or in batches) using some score function $g(S_\pi \cup S_l, \cdot) : S_u \to \mathbb{R}$ to score each point $x \in S_u$ and then the policy labels the point with the largest score. If $g$ were to be the first order difference of a submodular function $f$, i.e. $g(A, e) = f(A \cup \{e\}) - f(A)$, then the policy would be near-optimal. Moreover, as discussed above, intuitively we expect $g$ to have this property in most cases, since adding an extra point to a larger set usually has less effect than adding the same point to a smaller subset of the set.

The above motivates the use of LEASURE in active learning (Figure 2). In this experiment, the set $S$ is the Fashion-MNIST dataset consisting of greyscale images from one of 10 clothes classes (Xiao et al.

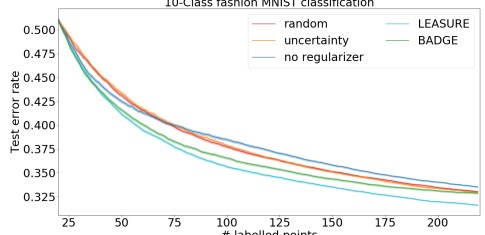

Figure 2: Combining submodular regularization with a learned active learning policy for 10-class Fashion-MNIST classification. The figure summarizes the classification error of a neural network trained on labelled images, as a function of the number of labelled images. Originally, random set of 20 images is selected, and then each policy greedily chooses the next image to label. The learned policies were trained on rollouts of length up to 30, and tested on rollouts of length 200. The "no regularizer" policy corresponds to Konyushkova et al. (2017), only in this case the features are parametrized by the neural network instead of being hand-engineered. "BADGE" corresponds to a sequential modification of (Ash et al., 2020). The results are averaged between 500 experiments, with standard error reported.

(2017)). The goal was to learn a policy that greedily selects "the best" point $x^* \in S_u$ to label, such that a neural network classifier trained on the labelled set $S_l \cup \{x^*\}$ produces the most accurate classification of the unlabelled images. In particular, we trained the above function $g$ to predict the

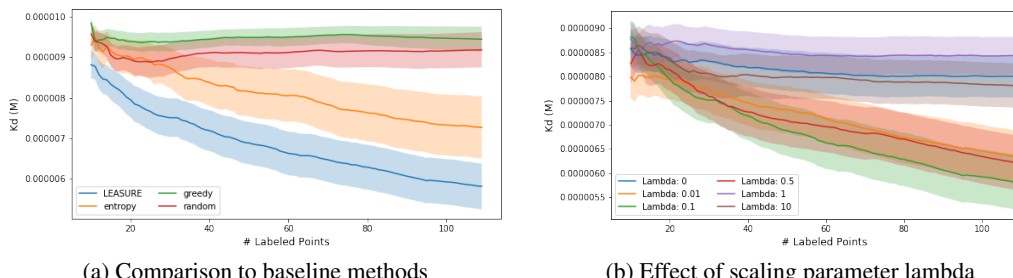

(a) Comparison to baseline methods   (b) Effect of scaling parameter lambda

Figure 3: Combining submodular regularization with a learned active learning policy for a protein engineering task. In (b), Lambda = 0 corresponds to the unregularized case. Error bars are plotted as standard error of the mean across 50 replicates.

accuracy gain $g^{\text{exp}}$ from labelling a point. The accuracy gain $g^{\text{exp}}$ was measured by training the neural network classifier on both $S_l$ and $S_l \cup \{x\}$ and then recording the difference in validation set classification accuracy. Since obtaining exact $g^{\text{exp}}$ for each datapoint is very expensive, we instead collected noisy labels $g_\epsilon^{\text{exp}} \approx g^{\text{exp}}$, obtained by training the classifier for only 10 epochs. The tuples $\{(S_l, x), g_\epsilon^{\text{exp}})\}$ were collected using DAgger with rollouts of length 30 (starting from a random batch of 20 images). For training, we used an initially unlabelled dataset with 60000 images, 2000 of which were set aside to use for evaluating validation accuracy. We trained two neural networks to approximate $g$ - an unregularized one, and one with a monotonicity and a submodularity regularizer (i.e. LEASURE). See Appendix B for details on architecture and training procedure.

The trained policies were tested on a set of 8000 images, with additional 2000 set aside for validation. At test time, we again started with a random batch of size 20 and then used each policy to sequentially select additional 200 images to label (Figure 2). The recorded test error rate was collected using real $g^{\text{exp}}$, i.e. a classifier trained until training loss reaches a certain threshold. The experiment was benchmarked against the "random" policy that randomly picked the next point, the "uncertainty" policy that selected the next point by maximizing uncertainty, the "no regularizer" policy that used DAgger with MSE loss, and "BADGE" from Ash et al. (2020). See Appendix B for details. Even though LEASURE was trained on much shorter rollouts using very noisy labels, it still outperformed all other baselines. This confirms our intuition that the submodular regularizer allowed the learned score function $g$ to find a local minima that generalizes well to out of sample.

### 6.3 PROTEIN ENGINEERING

By employing a large protein engineering database containing mutation-function data (Wang et al., 2019), we demonstrate that LEASURE enables the learning of an optimal policy for imitating expert design of protein sequences (see Appendix for detailed discussion of datasets). As in Liu et al. (2018) we construct a fully data-driven expert which evaluates via 1-step roll-out the effect of labeling each candidate data (in our case a protein mutant) with the objective of minimizing loss on a downstream regression task (predicting protein fitness).

When training the policy to emulate the algorithmic expert via imitation learning, we represent each state as two merged representations: (1) a fixed dimensional representation of the protein being considered (as the last dense layer of the network described in Appendix C), and (2) a similar fixed dimensional representation of the data already included in the training set (as a sum of their embeddings), including their average label value. At each step a random pool of data is drawn from the state space and the expert policy greedily selects a protein to label, which minimizes the expected regression loss on the downstream regression task (prediction of protein fitness). Once the complete pool of data has been evaluated, the states are stored along with their associated preference score, taken as their ability to reduce the loss in the 1-step roll out. Using these scores, the expert selects a protein sequence to add into the training set, and we retrain the model and use the updated model to predict a protein with the maximum fitness. This paired state action data is used to train the policy model at the end of each episode, as described in Liu et al. (2018). As we observe in Figure 3a, this method trains a policy which performs nearly identically to this 1-step oracle expert.

The use of submodular regularization enables the learning of a policy which generalizes to a fundamentally different protein engineering task. In our experiments, LEASURE is trained to emulate

a greedy oracle for maximizing the stability of protein G, a small bacterial protein used across a range of biotechnology applications (Sjbring et al., 1991). We evaluate our results by applying the trained policy to select data for the task of predicting antibody binding to a small molecule. As is the case with all protein fitness landscapes, the evaluation dataset is highly imbalanced, with the vast majority of mutants conferring no improvement at all. Because data is expensive to label in biological settings (proteins must be synthesized, purified and tested), we are often limited in how many labels can feasibly be generated, and the discriminative power among the best results is often more important than among the worst. To construct a metric with real-world applicability we assess each model by systemically examining the median Kd of the next ten data points selected at each budget, from 10 to 110 total labels. This method is utilized in recognisance of the extreme ruggedness of protein engineering landscapes, wherein the vast majority of labels are of null fitness, and the ability to select rare useful labels for the next experimental cycle is of key importance.

We observe that LEASURE outperforms all evaluated baselines, and that the inclusion of submodular optimization is mandatory to its success (Figure 3a). A greedy active learner which labels the antibody mutation with the best predicted Kd (the smallest) preforms approximately equivalently with selecting random labels. Use of dropout as an approximation of model uncertainty as in Gal and Ghahramani (2016) improves upon these baselines, although significant betterment is not achieved until approximately 35 labels are added. In comparison, the results from LEASURE diverge from all others nearly immediately, and the best model, which uses a lambda of 0.1, achieves a notable improvement in Kd, $5.81\,\mu$M, vs $7.27\,\mu$M achieved by entropy sampling. In support of methods success, we note that the learned policy preforms approximately as well as the greedy oracle which it emulates (Appendix Figure 7a). We observe that the results are robust within a range of possible lambda values (Figure Figure 3b and Appendix Figure 7b), and that without the use of submodular regularization the trained policy fails to learn a policy better than the selection of random labels. This is an important finding, as the method proposed by Liu et al. (2018) without LEASURE, has been shown to be a state-of-the-art method for imitation learning.

Based on these empirical results, LEASURE demonstrates significant potential as computational tool for *real-world automated experimental design tasks*: In particular, in the protein engineering task, LEASURE achieves the SOTA on the benchmark data-sets considered in this work. While LEASURE does involve repeated retraining of the protein engineering network, we observe that it returns strong results even with a single step of training. Additionally, the networks that are employed are very simple (Appendix C). This allows for reasonable training time (36 hours) and nearly instantaneous inference. Given the considerable time and cost of protein engineering, these computational budgets are quite modest. Protein engineering is a time consuming (months to years) and expensive undertaking (10's of thousands to millions of dollars). These projects usually strive to achieve the best possible results given a fixed budget. We have demonstrated in our work the ability deliver significant improvements in protein potency for the modest fixed budgets. Although the cost savings of engineering and testing an individual protein (or label) vary significantly based on the system, ranging tens to hundreds of dollars, we observe that to achieve a Kd of 8e-6 M LEASURE delivers an approximate cost savings of 65%, or 40 fewer labels than the next best method. The sequential synthesis and evaluation of each of these labels would likely span several months and additionally incur several thousands of dollars of materials costs.

## 7 CONCLUSION

In this paper, we introduce LEASURE, a data-driven decision making framework based on a novel submodular-regularized loss function. The algorithm was inspired by the recent developments of submodular-surrogate-based near-optimal algorithms for sequential decision making. We have demonstrated LEASURE on several diverse set of decision making tasks. Our results suggest that LEASURE can be easily integrated with modern deep imitation learning pipelines, and that it is efficient to run, while still reaching the best performance among the competing baselines. In addition to demonstrating the strong empirical performance on several use cases, we believe our work also provides useful insights in the design and analysis of novel information acquisition heuristics where traditional ad-hoc approaches are not feasible.

**Acknowledgements.** This research was supported in part by funding from NSF #1645832, NIH #T32GM112592, The Rosen Bioengineering Center, Raytheon, Beyond Limits, JPL, and UChicago CDAC via a JTFI AI + Science Grant. This work was additionally supported by NVIDIA corporation through the donation of the GPU hardware used in experiments.

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

# A  Proof for section 5

## A.1  Proof of Theorem 2

*Proof.* The high-level idea is to first connect the total expected utility of the learned policy $\hat{\pi}$ with the expected utility of the expert policy $\pi^{\mathrm{exp}}$, following the analysis in DAgger (Ross et al., 2011). Then, we will use the fact that $\pi^{\mathrm{exp}}$ is greedy with respect to $f$, an approximation to the submodular utility function $u$, to bound the one step gain of the $\pi^{\mathrm{exp}}$ against the $k$ step gain of running the optimal policy, and subsequently bound the total utility of the expert policy against the optimal policy. We would eventually obtain a similar result as Theorem 2, detailed as follows.

More concretely, following Theorem 3.4 in DAgger, we obtain that

$$\mathbb{E}[u(S_{\hat{\pi},k})] \geq \mathbb{E}[u(S_{\pi^{\mathrm{exp}},k})] - \Delta_{\max} k \hat{\epsilon}_N - O(1)$$

Here $\Delta_{\max}$ is the largest one-step deviation from $\pi^{\mathrm{exp}}$ that $\hat{\pi}$ can suffer. It is equivalent to the term $u$ in the DAgger paper. Since $f$ is $\epsilon$-close to a monotone submodular function $u$, we know that $\Delta_{\max} \leq \max_{A \subset \mathcal{V}, |A|=k} f(A) \leq \max_{A \subset \mathcal{V}, |A|=k} u(A) + \epsilon_E$, which is a constant once $u$ is given.

Next, since $\pi^{\mathrm{exp}}$ is greedily optimizing an $\epsilon_E$-approximation to a monotone submodular function $u$, we know that

$$\mathbb{E}[u(S_{\pi^{\mathrm{exp}},k})] \geq (1 - 1/e)\mathbb{E}[u(S_{\pi^*,k})] - k\epsilon_E$$

following the proof from Theorem 5 in (Chen et al., 2017b).

Combining both steps, we have that

$$\mathbb{E}[u(S_{\hat{\pi},k})] \geq (1 - 1/e)\mathbb{E}[u(S_{\pi^*,k})] - k(\epsilon_E + \Delta_{\max}\hat{\epsilon}_N) - O(1)$$

which completes the proof.

$\square$

## B  Suppelemental Details for the Set Cover and MNIST Active Learning Experiments

We provide additional results for the set cover experiments, under the same experimental setup as Figure 1a and 1b. The subplots 4a and 4b show the mean square error of learned policy $g$ as a function of the size of $S_l$. We provide a zoomed-in version of 4b in Figure 4c. In Figure 4c, we show it is clear that training the neural network on the monotonicity regularizer only does not help it learn out of sample - the error rapidly increases as soon as the test rollout length becomes larger than the training rollout length.

In Noisy Set Cover experiment (Figure 4a), each label of the element added to the superset was perturbed with $N(0, 1)$ noise. As a result, the variance of the total noise is linear in the number of sets. So, it is reasonable that the MSE error grows with number of sets - the policies cannot learn to predict random noise. While stochastic MSE of LEASURE and the no-regularizer policy are similar, LEASURE outperforms in the number of elements added, which is what matters in practice (Figure 1). These two figures confirm our intuition that when the problem is not exactly submodular, Leasure will still generalize better than no regularizer by learning to ignore small deviations from submodularity. Finally, it is also expected that DSF has a lower MSE than Leasure when the label noise is too large - Deep Submodular Functions are required to be submodular. When the stochasticity in the MSE becomes overwhelmingly large, that restrictive requirement becomes an advantage. However, when the MSE variance is not too large, the lack of expressiveness and the difficulty of optimization of DSF make it lose its advantage compared to Leasure.

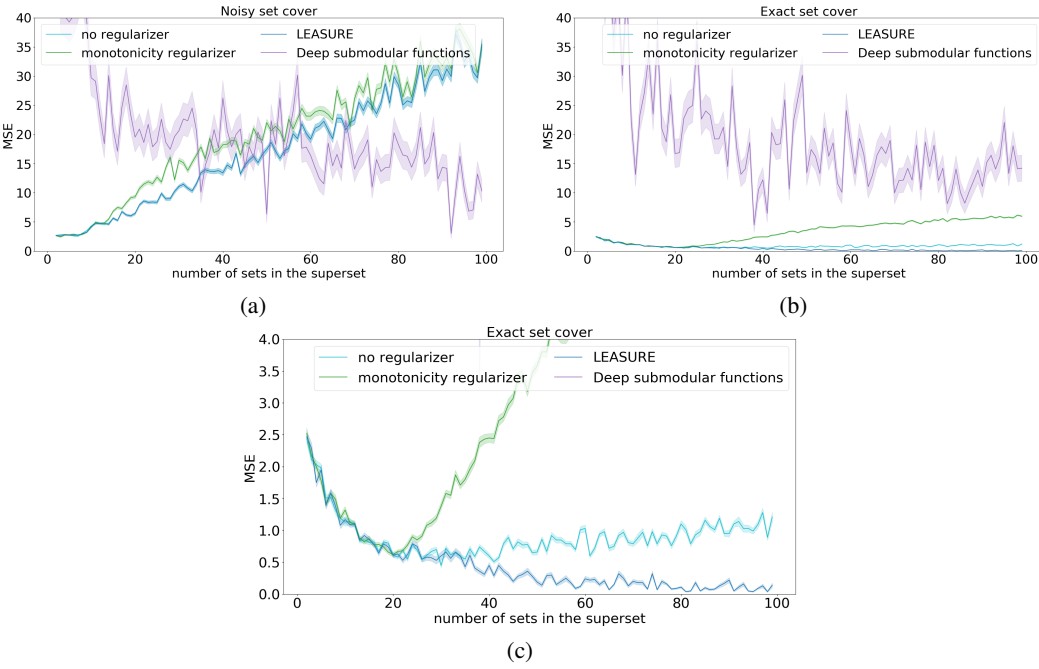

Figure 4: Supplemental results: Set cover

For completion, we also provide our architecture and parameter choices for both set cover and Learning Active Learning (LAL) on MNIST experiments. For set cover, the problem is too simple to require DAGGER (Ross et al., 2011). Instead, the tuples are generated randomly. For active learning on MNIST, the tuples are indeed generated using Algorithm 1. For MNIST, we first preprocessed our dataset with PCA, leaving the number of vectors necessary to achieve 80% covariance on the training set (24 vectors). That was necessary to allow the comparison with DSF. For set cover, each element was a set $v$ containing 23 elements $v^1, v^2, .., v^{23}$, where $v^i$ was an integer corresponding to the label of the species. As a neural network input, $v$ was simply represented as a vector of $[v^1, ..., v^{23}]$.

Both set cover and MNIST used a modified Deepset architecture (Zaheer et al., 2017) for score networks as follows: Given a set $A = \{v_0, ..., v_k\} \subset V$ and a datapoint $v \in V$, the score network $g$ first preprocesses all inputs $v_0, ..., v_k, v$ to obtain learned embeddings $\bar{v}_0, .., \bar{v}_k, \bar{v}$. (See Figure 5) Then, the elements in $A$ are combined using Deepsets architecture to produce a learned set embedding $\bar{A}$. Finally, $\bar{A}$ and $\bar{v}$ are concatenated and then a learned linear layer and a Leaky ReLu nonlinearity are applied to produce $g(A, v)$. (See Figure 6). All dense layers have 64 neurons and a bias term. Using this Deepsets-like framework, we achieve permutation invariance of elements in set $A$ while also keeping the network expressive enough to learn a wide range of functions.

### Learning element representation

Input $v \in \mathcal{V}$ $\longrightarrow$ Dense Layer + Tanh $\longrightarrow$ Dense Layer + Tanh $\longrightarrow$ Embedding $\bar{v}$

Figure 5: Score neural network architecture illustration

### Combining element representation using DeepSets

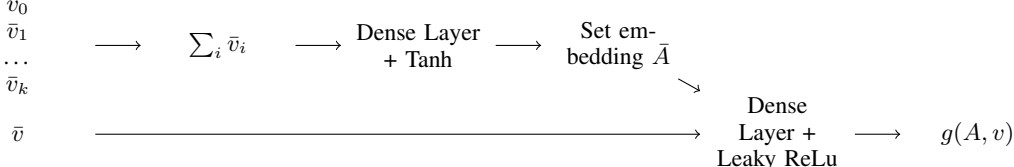

Figure 6: Score neural network architecture illustration.

For both tasks, the score networks are trained using ADAM with a learning rate of 1e-3. Beta parameter from Line 2 in LEASURE was picked randomly to be $\frac{4}{5}$. From experiments, the exact value of the parameter did not matter as long as it starts with at least $\frac{1}{2}$ and degrades towards almost 0 after $N$ iterations. The $\lambda$ and $\gamma$ parameters were picked using a hyperparameter sweep in log space. As per our intuition, we have found that the strength of the parameters should reflect your certainty that the task is submodular and/or monotone. For set cover, $\lambda = 0.1, \gamma = 0.5$, while for active learning $\lambda = 0.001$ and $\gamma = 0.001$. Notice that the values are not comparable between different experiments: for MNIST Learning Active Learning (LAL), $g^{\text{exp}}(A, v) \in [0, 1)$ outputs the accuracy gain of adding $v$ to $A$ and training a supervised model on it; for set cover, $g^{\text{exp}}(A, v) \in \{0, 1\}$ outputs the number of new elements added to the set by adding $x$ to $A$. For LAL, the values of $g^{\text{exp}}$ are usually much smaller than 1, particularly for larger sets. Thus, the values for the two regularizers had to be smaller so that the model learns not just the regularizer.

Finally, we wanted to discuss our baselines in Fashion MNIST experiments. In Figure 2, we have four baselines: random, uncertainty, BADGE (Ash et al., 2020), and no regularizer. The no regularizer baseline was trained identically to LEASURE, except for the absense of submodularity and monotonicity regularizers. The no regularizer baseline performed well on the sets with up to 30 additional points - corresponding exactly to the length of the training rollouts. However, it failed to generalize. On the other hand, the submodular regularizer allowed the learned score function to find a local minima that generalized well to out of sample. Finally, BADGE did not seem to perform well when the number of datapoints in the set was large, likely because the gradient signal from adding any one additional datapoint was too weak and thus the selection of the next best datapoint was too noisy.

Some more details regarding BADGE (Ash et al., 2020). The authors do not learn a policy, instead, they use gradients of the classifier (gradient embedding) to select a useful, diverse batch. Although BADGE was originally made for a batch setting, the authors' main idea is still applicable to our case: they argued that the next datapoint(s) can be selected by looking at which fictitious labels would produce the largest gradients in the classifier network. Therefore, we replaced the kmeans++ algorithm the authors suggested with simply selecting the datapoint that corresponds to the largest gradient norm. This algorithm has an advantage that it does not require a trained policy network. However, it provides no guarantees about submodularity of the resulting policy, and, in our experiments, the performance degrades with the size of the set - likely because the gradient signal from

adding any one additional datapoint was too weak and thus the selection of the next best datapoint was too noisy. Since BADGE requires a neural network classifier/regressor, we could not use it as a baseline for Set Cover (Set Cover regression function is simply adding all elements in the superset).

The no-regularizer baseline is similar to that of Konyushkova et al. (2017). However, the problem considered in Konyushkova et al. (2017) is not compatible with most of the tasks we considered here (for MNIST, yes if we use random forest classifiers; but for others not). Furthermore, Konyushkova et al. (2017) treated the problem under a classical supervised learning setting this is often not desirable, given that we are learning a policy from non i.i.d. data samples.

## C    Supplemental Details for the Protein Engineering Experiments

**Dataset**    Our datasets were identified in Protabank (Wang et al., 2019) for training of active learning policies and benchmarking of performance. In selecting datasets upon which to train our active learning models several factors were considered. As the state space of possible protein variants for typical engineering application is very large, size is our foremost criteria. Additionally it will be advantageous to use datasets which characterize mutations to all amino acids (as opposed to Alanine scans), and those which include epistatic interactions. We also desire to identify datasets which have a high quality, quantitative readout, such as calorimetry, fluorescence, or SPR data.

**Protein Engineering Methods**    Embeddings of protein sequences were created using the TAPE repository (Rao et al., 2019) according to the UniRep system as first proposed in Alley et al. (2019). UniRep produces protein embeddings as a matrix of shape (length protein sequence, 1900), although we average together the embeddings only of positions being engineered to produce a consistent embedding of shape (1900,). We have implemented the active learning imitation learning algorithm proposed in Liu et al. (2018) to work with the protein embedding representations described above. Pseudocode for this method is presented in Algorithms 1 and 2 from the original work. As in Liu et al. (2018), our policy network consists of a single dense unit which acts sequentially on the pool of samples being considered to produce a preference score. Our downstream protein engineering network (which was used to compute the preference score of the expert policy) acts on the protein embeddings prepared using TAPE. The network consists of an attention layer, followed by a 1-dimensional convolution layer (128 filters, kernel size 3), before being flattened and applying two fully connected layers of 128 units each. When predicting protein fitness, dropout is applied with a probability of 0.5 and an additional dense layer is applied with one unit and linear activation. Both networks are trained using ADAM with a learning rate of 1e-3. The implementation of this part of the project is nearly identical to Liu et al. (2018), only changing the data representation, protein fitness network structure, and values of K (30), B (100) and T (20) as listed in the appendix of our work. Beta is fixed at 0.5, although the method was shown to be robust to a range of values. At training time, 100 labels are randomly selected for evaluating the effect of the greedy oracle, and 10 data are randomly selected to form the initial data set for learning. The superset is appended at each step of training the policy to maintain a size of 2x the labeled dataset. The training of a policy using these settings takes 36 hours on a modern multiprocessor computer equipped with an NVIDIA Titan V GPU.

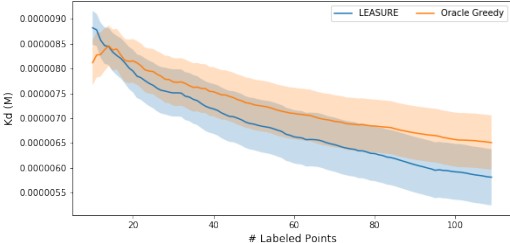    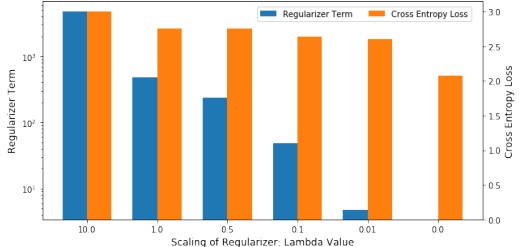

(a) Comparison of policy to greedy oracle which it emulates

(b) Effect of scaling parameter lambda and empirical evidence for selecting its value

Figure 7: Supplemental results for the protein engineering experiments of Section 6.3: (a) We observe that the policy learned by LEASURE preforms approximately as well as the greedy oracle which it emulates. In this experiment the policy was derived from the training set, but the greedy oracle is operating on the test set. (b) Lambda linearly scales the value of the regularizer term. When lambda takes value 0.01, the magnitude of the (scaled) regularizer term (represented by the blue bar) aligns the best with the magnitude of the cross entropy loss (represented by the orange bar). This is consistent with what we observed in Figure 3b where $\lambda = 0.01$ leads to well-regularized model behavior.

