# OpenReview forum: "Learning to Make Decisions via Submodular Regularization"
_ICLR.cc/2021/Conference — ICLR 2021 Poster_

### Official Review · AnonReviewer4 · 2020-10-19
**Needs more details on experiments**

**Rating:** 6
**Confidence:** 4

**Review:**

1. Summary of the paper contributions
    (1) This paper proposes a 'submodular-norm' based regularization term when learning a score function based on data-driven optimization, which encourages the underlying utility function to exhibit "diminishing return". The learned scoring function can be used as a greedy heuristic in combinatorial sequential decision making tasks, in which expert policy is expensive to evaluate.
    (2) The proposed approach can be easily integrated with other imitation learning pipelines when the considered problems fall into the (approximate) submodular set. This is shown in the experiment section by examples including set cover, learning active learning,  and learning to imitate expert design in protein engineering.

2. Strong and weak points of the paper
    (1) Strong points: The considered problem is interesting and meaningful. This paper provides a new 'submodular-norm' when learning the scoring function, which seems to be effective in the shown examples. Since the proposed algorithm is a specialization of  the DAgger algorithm (Ross et al., 2011), the performance guarantee for the learned scoring function is established.
    (2) Weak points: Although the proposed algorithm can achieve the desired performance guarantees in Theorem 1 and 2, these results are a natural extension (or direct result) of prior work Ross et al., 2011. The proposed 'submodular-norm' seems to be effective in the shown 3 examples, but many details are omitted. This makes it hard to tell the proposed algorithm's effectiveness and efficiency in practice.

3. My recommendation
    I tend to reject this paper, but am open to change my score after rebuttal. My main concern is on the experiment part, which lack many details and baseline methods.

4. Supporting arguments for your recommendation
    The paper's theoretical results are a natural extension or direct result of prior work, so I focus more on the empirical performance part such as whether the propose algorithm is efficient to train and the effectiveness compared to baseline methods. Although the results shown in the paper seem to be effective, many details and baseline methods are not shown. Also, the second example of learning active learning policy seems to be not very efficient. Because it needs a (noisy) expert scoring function value for every evaluated data point. This 'expert' scoring function is trained for every data point encountered in training. Although it uses only 10 epochs to obtain this value, it is still very time-consuming considering the rollout length and the total number of rounds.

5. Questions
    (1) For the m shown in Theorem 1, is it the T in Algorithm 1 step 7?
    (2) What does the k stand for in Theorem 2?
    (3) For the 3 examples, can the author/s provide details on how the labeled tuples are generated? Is it by following Algorithm 1? If so, what's the \beta and N values for these examples?
    (4) For set cover example, the compared DSF generalizes better to larger rollout lengths as shown in Fig. 1. Does it mean the DSF may achieve the total of 119 species with least number of rollout sets in the superset? Any result in this metric?
    (5) For the 3 examples, the estimated scoring function g is parametrized by neural network. Can the author/s provide details on how the parametrization is constructed?
    (6) Why does the author/s not include other imitation learning methods as baseline like Liu et al., 2018 discussed in Section 2?
    (7) How to determine the regularization parameters \lambda and \gamma? For other compared methods, did the author/s tune their hyper-parameters?
    (8) The 'no regularizer' policy in Fig. 2 is adapted from Konyushkova et al., 2017 without hand-engineered features. Will this lower the method of Konyushkova et al., 2017's performance in general?

---

> ### Author Response · Authors · 2020-11-23
> **Authors' response to AnonReviewer4 (Clarifications to questions on algorithmic details, efficiency, baselines, and analysis)**
>
> Thank you for the comments! Below please find our response (**A**) to the main questions (**Q1-7**).
>
> **Q1: [Algorithmic details for neural network configuration on the three tasks]**
> *(“Can the author/s provide details on how the parametrization is constructed?”)*
>
> **A**: Thanks for the suggestions of adding necessary algorithmic details! Since the question on the neural network configurations is also raised in other reviews, please refer to our answer to **Q1** in our response to *AnonReviewer2*.
>
> We will modify our experimental section and appendix in the revision accordingly, to include more information about the training procedure and the neural network architecture that were used for the three tasks.
>
> Please let us know if our above feedback addresses your concerns.
>
>
> **Q2: [Further algorithmic details]**
> *(“How to determine the regularization parameters \lambda and \gamma? For other compared methods, did the author/s tune their hyper-parameters?” )*
>
> **A**:  The \lambda and \gamma parameters were picked using a hyperparameter sweep in log space. As per our intuition, we have found that the strength of the parameters should reflect your certainty that the task is submodular and/or monotone:
> For set cover, lambda=0.1, gamma=0.5
> For active learning, lambda=0.01, gamma=0.001.
>
> Notice that the values are not comparable between different experiments:
> For MNIST Learning Active Learning (LAL), $g^{\exp} (A,v) \in [0,1)$ outputs the accuracy gain of adding v to A and training a supervised model on it;
> For set cover, $g^{\exp}(A,v) \in {0,1}$ outputs the number of new elements added to the set by adding x to A. For LAL, the values of $g^{\exp}$ are usually much smaller than 1, particularly for larger sets. Thus, the values for the two regularizers had to be smaller so that the model learns not just the regularizer.
>
>
> **Q3: [Clarification on how labeled tuples are generated]**
> *(“For the 3 examples, can the author/s provide details on how the labeled tuples are generated? Is it by following Algorithm 1? If so, what's the \beta and N values for these examples?”)*
>
> **A**: For set cover, the problem is too simple to require DAgger. Instead, the tuples are generated randomly. For active learning on MNIST, the tuples are indeed generated using Algorithm 1. The detailed description is given in the experiment section and appendix B. For active learning on MNIST, beta= ⅘, and N=20; For the protein engineering task, we use beta=½.
>
> **Q4: [Efficiency of LEASURE policy for active learning (MNIST)]**
> *(“the second example of learning active learning policy seems to be not very efficient. Because it needs a (noisy) expert scoring function value for every evaluated data point.., it is still very time-consuming considering the rollout length...”)*
>
> **A**: The main goal of our paper was to show that an expensive oracle can be replaced with an inexpensive, data-driven greedy policy at *test time*. We specifically picked experiments with an expensive oracle to showcase the advantages of our idea.
>
> Moreover, we would like to highlight our empirical observation that running LeaSuRe with a really noisy oracle still achieves good performance.
>
>
> **Q5: [Performance against  Konyushkova et al., 2017]**
> *(The 'no regularizer' policy in Fig. 2 is adapted from Konyushkova et al., 2017 without hand-engineered features. Will this lower the method of Konyushkova et al., 2017's performance in general?)*
>
> **A** : The problem considered in Konyushkova et al., (2017) is not compatible with most of the tasks we considered here (for MNIST, yes if we use random forest classifiers; but for others not). Furthermore, Konyushkova et al (NeurIPS17) treated the problem under a classical supervised learning setting — this is often not desirable, given that we are learning a policy from non i.i.d. data samples.
>
>
> **Q6: [Clarification on the imitation learning baseline]**
> *(Why does the author/s not include other imitation learning methods as baseline like Liu et al., 2018 discussed in Section 2?)*
>
> **A**: Thank you for emphasizing the existing imitation learning baselines. In fact, **we have already included the baseline** in our submission---In figure 3b, when lambda is set to zero, our algorithm is exactly that of Liu et al 2018. We observe that the method fails to learn an effective policy. We also considered adding PAL, as proposed by Fang (https://arxiv.org/abs/1708.02383), but ultimately did not, as it is shown to be inferior to Liu’s work, and would require approximately 350x more compute time. Our method trains in approximately 36 hours, making this an unachievable scale-up.
>
> We will revise our discussion in baselines and further clarify this in the revision.
>
>
> **Q7: [Clarification on Theorem 2]**
> (“What does the k stand for in Theorem 2?”)
>
> **A**: $k$ is the total number of elements the policy $\hat{\pi}$ is going to select. In experiments, $k$ is the total number of labelled training data the learned policy can query.

---

> > ### Comment · AnonReviewer4 · 2020-11-24
> > **Feedback after rebuttal**
> >
> > Thanks the author/s for the helpful response.
> > One of my questions is not answered yet, so I re-post it here again with a follow-up question.
> > (4) For set cover example, the compared DSF generalizes better to larger rollout lengths as shown in Fig. 1. Does it mean the DSF may achieve the total of 119 species with least number of rollout sets in the superset? Any result in this metric?
> >
> > (follow-up)  Based on my question Q3: [Clarification on how labeled tuples are generated], are the related parameters also determined by hyper-parameter sweep in log space?
> >
> > The details provided by the author/s are really useful. Could these details along with the response to Reviewer #2 to be included in the final submission?

---

> > > ### Author Response · Authors · 2020-11-25
> > > **Authors' response to the follow-up questions of AnonReviewer4**
> > >
> > > > For set cover example, the compared DSF generalizes better to larger rollout lengths as shown in Fig. 1. Does it mean the DSF may achieve the total of 119 species with least number of rollout sets in the superset? Any result in this metric?
> > >
> > > We apologize for missing this question in our initial response. Thank you for reminding us of this question---this is a great point to clarify. We tested our approach and DSF following the proposed metric. We ran the two experiments over 100 random trials, each up to 300 elements in the superset. Out of a 100 runs, 41 LeaSuRe experiments collected 119 species, while only 1 DSF experiment did. Moreover, out of the experiments that collected all the species, the average number of sets was 135 for LeaSuRe, versus 223 for DSF.
> > >
> > > We ran the experiments up to 300 elements as DSF takes a prohibitively long time to run.
> > >
> > >
> > > > [Clarification on how labeled tuples are generated], are the related parameters also determined by hyper-parameter sweep in log space?
> > >
> > > While we have tried a few different values for $\beta$, we have found that it has little to no effect on the algorithm performance, except when beta was zero. So these specific choices of $\beta$ were picked randomly (Set Cover, MNIST Learning Active Learning) or according to prior work (in protein engineering, $\beta$ was selected to be 0.5 as in Liu et al.)
> > >
> > >
> > > > Could these details along with the response to Reviewer #2 to be included in the final submission?
> > >
> > > Yes! We have incorporated the updates in the rebuttal revision. We will upload the rebuttal revision shortly and summarize the changes in a separate response.

---

### Official Review · AnonReviewer2 · 2020-10-28
**Potentially useful idea, but the paper is lacking details**

**Rating:** 7
**Confidence:** 4

**Review:**

Authors propose to approximate the marginal gain of the submodular function (also called discrete derivative) with a neural network. The data for training the model is collected with DAgger, and therefore the proposed method inherits guarantees of DAgger.

Clarity: My main concern is that some of the important details are not discussed in the paper. Authors only mention that $g$, the marginal gain approximator, is a neural network; I couldn't find any discussions on what kind of neural network was used. Since the input of the neural network includes a power set, the input representation of the data as well as the neural network architecture would be of interest. Also, the description of protein engineering experiment is a bit confusing. It mentions that the expert is trained on a downstream regression task, but it is also mentioned that 'LeaSure is trained to emulate a greedy oracle for maximizing the stability of protein G', which is inconsistent. Even in the Appendix C, details on the experiments such as the description of parameters or the pseudocode of the algorithm are deferred to references.

Quality: The main claim of the paper is that the cost of evaluating the expert policy can be reduced by approximating it as a neural network. However, there are many factors contributing the cost of employing the proposed method, as a potentially large neural network would need to be trained, and data points for training the network should be acquired. A larger neural network would make it more expensive to evaluate $g$, diminishing the value of approximating the expert. Also, a larger neural network may be less sample-efficient. It would be interesting to discuss how these other cost factors should be considered in order to optimize the total cost.

Originality, Significance: The idea of approximating the marginal gain with a neural network seems new, and also broadly applicable across applications of submodular optimization.

Pros:
* The idea is original and creative
* The proposed method is widely applicable

Cons:
* The practical utility is quite uncertain and questionable
* The paper lacks important details, even the architecture of neural networks used

---

In the revision, authors provided detailed clarifications on their approach and practical utility of the proposed method. Therefore, I am changing the rating of my review.

---

> ### Author Response · Authors · 2020-11-23
> **Authors' response to AnonReviewer2 & AnonReviewer4 (Clarifications to algorithmic details and significance of this work)**
>
> Thank you for the comments! Below please find our response (**A**) to the main questions (**Q1-3**).
>
> **Q1: [Algorithmic details for the three tasks]**
> *(“The paper lacks important details, even the NN architecture”)*
>
> **A**:  Thanks for raising this issue. In the following, we provide a detailed description of our experimental setup for each of the tasks/datasets---including the neural networks used, parameter configuration, etc.
>
> Briefly, as the main goal of our paper was to showcase the performance of a novel submodular regularizer, we simply picked basic network architectures (in case of set cover and MNIST active learning) or in accordance with prior work (protein engineering task):
>
> - For set cover and MNIST active learning, we used a modified Deepset architecture to preserve permutation invariance. Both policy networks had a 4-layer dense neural network with hidden layers consisting of 64 neurons and tanh nonlinearities and the output layer with Leaky Relu nonlinearity. Both networks are trained using ADAM with a learning rate of 1e-3. Beta parameter from Line 2 in Leasure was picked randomly to be ⅘.
>
> - For the protein engineering task, our policy network consists of a single dense unit which acts sequentially on the pool of samples being considered to produce a preference score, exactly as implemented in Liu et. al 2018. Our downstream protein engineering network (which was used to compute the preference score of the expert policy) acts on the protein embeddings prepared using TAPE (the UniRep setting, with slight modifications as mentioned in appendix C). The network consists of an attention layer, followed by a 1-d convolution layer (128 filters, kernel size 3), before being flattened and applying two fully connected layers of 128 units each. When predicting protein fitness, 50% dropout and an additional dense layer is applied with one unit and linear activation. Both networks are trained using ADAM with a learning rate of 1e-3. The implementation of this part of the project is nearly identical to Liu, only changing the data representation, protein fitness network structure, and values of K, B and T as listed in the appendix of our work.
>
>
> **Q2: [Clarification on the algorithmic details of the expert policy for the protein engineering experiment ]**
> *(“It mentions that the expert is trained on a... regression task, but it is also mentioned that 'LeaSure is trained… for maximizing the stability of protein G'...”)*
>
> **A**: Below please find our clarification of the expert policy and the training procedure:
>
> - At each step, the expert policy greedily selects a protein to label, which minimizes the expected regression loss on the downstream regression task. Once the expert policy selects a protein sequence, we retrain the model and use the updated model to predict a protein with the maximal stability.
>
> - When training the greedy policy using imitation learning, we represent each state via (1) a fixed dimensional representation of the protein being considered (described above in the network details), and (2) a fixed dimensional representation of the data already included in the training set (as a sum of their embeddings), including their average label value. Once the complete pool of data has been evaluated, the states are stored along with their associated preference score -- their ability to reduce the loss in the one-step roll out. This paired state action data is used to train the policy at the end of each episode, as described in Liu et al., (2018). As we observe in Figure 5a, this method trains a policy with performance nearly identical to the greedy oracle.
>
>
> **Q3** [Clarification on the significance of this work]
> *(“The practical utility is quite uncertain and questionable”)*
>
> **A**:  We would like to highlight that our empirical results have demonstrated LeaSuRe to be an extremely promising computational tool for **real-world automated experimental design tasks**: In particular, for the protein engineering task, LeaSuRe achieves the SOTA on the benchmark datasets considered in this work. **These experiments are conducted as part of a collaboration with real protein biologists, and this method will be incorporated into their design workflow**.
>
> While LeaSuRe does involve repeated retraining of the protein engineering network, we observe that it returns strong results even with a single step of training. The networks that are employed are *quite simple*, as described in our response to **Q1**. This allows for reasonable training time (36 hours) and nearly instantaneous inference. Given the considerable time and cost of protein engineering, these computational budgets are quite modest. It often can cost $100-1000 and several weeks to prepare a protein variant, depending on the system being studied and available resources.  Therefore, with the novel submodular-norm loss function, LeaSuRe already achieves orders of magnitudes of saving in both the computational and monetary costs.

---

> > ### Comment · AnonReviewer2 · 2020-11-24
> > **thanks for clarifications**
> >
> > It's great to hear that the protein engineering example is already a practical application. This was difficult to appreciate with the original draft, because I had trouble following the description of the experiment.
> >
> > > At each step, the expert policy greedily selects a protein to label, which minimizes the expected regression loss on the downstream regression task. Once the expert policy selects a protein sequence, we retrain the model and use the updated model to predict a protein with the maximal stability.
> >
> > So this "prediction of a protein with the maximal stability" seems to be the real down-stream task? Can't we use this task instead of the regression task for training of the model?
> >
> > > (from the paper:) To construct a metric with real-world applicability we assess each model by systemically examining the median Kd of the next ten data points selected at each budget, from 10 to 110 total labels.
> >
> > This measures the stability of proteins? I don't think the motivation behind this metric was explained.
> >
> > > (from the paper:) we note that the learned policy preforms approximately as well as the greedy oracle which it emulates
> >
> > Figure 3(a) and 5(b) seems to suggest that Leasure actually "outperforms" greedy "oracle" quite a bit? Is this what we should be expecting?
> >
> > >  It often can cost $100-1000 and several weeks to prepare a protein variant, depending on the system being studied and available resources.
> >
> > This is a good argument. It would be even better if we can convert this to a metric better understandable by any ML researcher. Something along the lines of (sorry for my naivety, my understanding on this experiment is still poor, as evidenced above): "In order to find a functional protein, it's stability should be below XXX Kd... With the baseline method it will require XXX labels (approx \\$ YYY) to achieve this level of Kd... in contrast, LeaSure allows us to achieve the same level of Kd with AAA labels in (approx \\$ ZZZ)."

---

> > > ### Author Response · Authors · 2020-11-25
> > > **Authors' response to the follow-up questions of AnonReviewer2**
> > >
> > > > It's great to hear that the protein engineering example is already a practical application. This was difficult to appreciate with the original draft, because I had trouble following the description of the experiment.
> > >
> > > Thank you for your constructive feedback. It has helped to guide us in improving the papers completeness and clarity.
> > >
> > > > So this "prediction of a protein with the maximal stability" seems to be the real down-stream task? Can't we use this task instead of the regression task for training of the model?
> > >
> > > We apologize for the confusion in our language. The regression task is indeed one and the same as the prediction of the protein with the maximum stability. The language has been modified in this section of the paper to convey this.
> > >
> > > > This measures the stability of proteins? I don't think the motivation behind this metric was explained.
> > >
> > > This method is utilized in recognisance of the extreme ruggedness of protein engineering landscapes, wherein the vast majority of labels are of null fitness, and the ability to select rare useful labels for the next experimental cycle is of key importance.
> > >
> > > > Figure 3(a) and 5(b) seems to suggest that LeaSuRe actually "outperforms" greedy "oracle" quite a bit? Is this what we should be expecting?
> > >
> > > The greedy metric in 3(a) is not the greedy oracle, but rather a greedy active learner, that predicts the best label to add using the current model weights. The greedy oracle is demonstrated in figure 7(a), and is closely aligned with LeaSuRe (error bars significantly overlap).
> > >
> > > > It would be even better if we can convert this to a metric better understandable by any ML researcher.
> > >
> > > Thank you for this suggestion. We have amended the manuscript to indicate that while projects are seldom governed by achieving a target potency, but rather achieving the best possible results given a fixed budget, we are able to deliver an approximate cost savings of 65\%, or 40 fewer labels than the next best method in our experiment to achieve a set Kd of 8e-6 M. Given that the cost of engineering and testing individual proteins vary significantly based on the system being engineered, ranging tens to hundreds of dollars, this is a potentially dramatic amount.

---

### Official Review · AnonReviewer3 · 2020-10-28
**Nice idea, but some questions**

**Rating:** 7
**Confidence:** 3

**Review:**

This paper combines combines submodular surrogates for sequential decision making with imitation learning. Specifically, it proposes to learn an acquisition function g by imitating an expert which is assumed to be following a greedy policy wrt a general submodular surrogate f. This is accomplished by regularizing g to encourage diminishing returns and monotonicity. The learning algorithm is a modified version of DAgger which is consistent with the expert and provably near-optimal utility. Results outperform baselines on various sequential decision making tasks.


Novelty/Impact
- Elegant, novel combination of DAgger with submodular regularization
- Potentially impactful idea (comparison to additional baselines would confirm this)
- Theoretical contribution is a straightforward combination of (Ross, 2011) and (Golovin and Krause, 2011). On its own this is not very impactful

Experiments:
- Good, concise experiments across a range of applications.
- The expert's utility function is shown to exhibit submodularity, strengthening the validity of the proposed approach
- Baselines are illustrative, but the paper would benefit from having additional baselines (for example [1-2] for active learning)

Clarity/Correctness:
- In the proof of Theorem 2, can you explain the first inequality when bounding the one-step gain of $\hat{\pi}$? What if $\pi^{exp}$ is not along the learned trajectory?
- Do the baselines in section 6.1 use Algorithm 1 (i.e. DAgger) with different loss functions?
- typo: "groundset" should be "ground set"

Pros
- Good organization
- Potentially high impact
- Thorough experimental methodology

Cons
- Some statements unclear
- Somewhat weak baselines
- Limited theoretical contribution

Questions:
- Is the bound in Theorem 2 tight?
- How does the algorithm perform when the surrogate is non-monotone?
- Is the poor performance on noisy set cover due to noisy data or submodularity being violated?


[1] Wei, Iyer, and Bilmes. Submodularity in Data Subset Selection and Active Learning, ICML 2015. http://proceedings.mlr.press/v37/wei15.html

[2] BADGE Ash et al. Deep Batch Active Learning by Diverse, Uncertain Gradient Lower Bounds, ICLR 2020. https://arxiv.org/abs/1906.03671


EDIT: The authors addressed my main concerns, fixed a crucial bug in the claim/proof of Theorem 2, and added many more clarifying details to a revised submission. Therefore I will keep my rating the same

---

> ### Author Response · Authors · 2020-11-19
> **Authors' response to AnonReviewer3 (Clarifications to questions in proofs and empirical performances)**
>
> Thank you for the detailed review and comments. Below please find our response (**A**) to the main questions (**Q1-4**).
>
> **Q1: [Clarification of Thm 2 proof]**
> *(“In the proof of Theorem 2, can you explain the first inequality when bounding the one-step gain of \hat{π}? What if π^{exp} is not along the learned trajectory?”)*
>
> **A**: Thanks for raising this question! We agree that the current proof of Theorem 2, in particular, the inequality bounding the one-step gain of the *learned policy* is flawed---we provide an easy fix for this issue below.
>
> The high-level idea is to first connect the total expected utility of the learned policy with the expected utility of the expert policy $\pi^{\exp}$, following the analysis in DAgger [1]. Then, we will use the fact that $\pi^{\exp}$ is greedy with respect to $f$, an approximation to the submodular function $u$, to bound the one step gain of the $\pi^{\exp}$ against the $k$ step gain of running the optimal policy, and subsequently bound the total utility of the expert policy against the optimal policy. We would eventually obtain a similar result as Theorem 2, detailed as follows.
>
> More concretely, following Theorem 3.4 in DAgger, we obtain that
> $$\mathbb{E}[u(S_{\hat{\pi}, k})] \ge  \mathbb{E}[u(S_{\pi^{\exp}, k})] - \Delta_{\max}k\hat{\epsilon}_N - O(1)$$
>
> Here $\Delta_{\max} $ is the largest one-step deviation from $\pi^{\exp}$ that $\hat{\pi}$ can suffer. Since $f$ is $\epsilon$-close to a monotone submodular function $u$, we know that $\Delta_{\max} \le \max_{A \subset \mathcal{V}, |A|=k} f(A) \le \max_{A \subset \mathcal{V}, |A|=k} u(A) + \epsilon_E$, which is a constant once $u$ is given.
>
> Next, since $\pi^{\exp}$ is greedily optimizing an $\epsilon_E$-approximation to a monotone submodular function $u$, we know that
> $$\mathbb{E}[u(S_{\pi^{\exp}, k})] \ge  (1-1/e)\mathbb{E}[u(S_{\pi^*, k})] - k\epsilon_E$$ following the proof from Theorem 5 in [2].
>
> Combining both steps, we have that
> $$\mathbb{E}[u(S_{\hat{\pi}, k})]  \ge  (1-1/e)\mathbb{E}[u(S_{\pi^*, k})] - k(\epsilon_E + \Delta_{\max}\hat{\epsilon}_N) - O(1)$$
>
> We will incorporate the updated Theorem 2, and update the full proof of Theorem 2 accordingly in the (rebuttal) revision.
>
> Does this address your concern in the soundness of our results?
>
>
> [1]: Ross, Stéphane, et al.  "A reduction of imitation learning and structured prediction to no-regret online learning." Proceedings of the fourteenth international conference on artificial intelligence and statistics. 2011.
> [2]: Chen, Yuxin, et al. "Efficient Online Learning for Optimizing Value of Information: Theory and Application to Interactive Troubleshooting." Proceedings of the 33rd Conference on Uncertainty in Artificial Intelligence (UAI 2017). Vol. 2. Curran Associates, Inc., 2017.
>
> **Q2: [Further clarification on Theorem 2]**
> *(“Is the bound in Theorem 2 tight?”)*
>
> **A**:  The above updated result of Theorem 2 provides a lower bound of the expected utility of the learned policy. Here we do not claim the bound to be tight, as the intermediate steps (i.e., the lower bound for approximate submodularity maximization (with additive noise) and the generalization bound involving DAgger) could be tightened.
>
>
> **Q3: [Effect of (non-)monotonicity on the performance]**
> *(“How does the algorithm perform when the surrogate is non-monotone”)*
>
> **A**: Our theoretical justification of the proposed algorithm assumes that the greedy heuristic follows a monotone, approximately submodular surrogate function. We observe that under settings where the objective of the expert policy is not (strictly) monotone by construction---e.g., the *noisy set cover* task with perturbed coverage function, and the *active learning* task with a greedy expert maximizing the expected error reduction---LeaSuRe **still significantly outperforms the baselines**. Our interpretation is that LeaSuRe encourages the learning algorithm to construct an (approximately) monotone submodular surrogate function, while being aligned with observed utility gain of a greedy expert (despite the expert utility may not be exactly monotone submodular).
>
>
> **Q4: [Performance of LeaSuRe on noisy set cover]**
> *(“Is the poor performance on noisy set cover due to noisy data or submodularity being violated?”)*
>
> **A**:  We would like to clarify that LeaSuRe actually *outperforms the baselines by a great margin*, on both the noisy set cover and the exact set cover tasks. This in fact highlights the robustness of LeaSuRe under mild violation of submodularity.
>
> Is it possible that you were referring to the poor performance of the DSF baseline, rather than LeaSuRe?

---

> > ### Author Response · Authors · 2020-11-19
> > **Authors' response in additional baselines**
> >
> > **Q5: [Additional Baselines]**
> > *(“Baselines are illustrative, but the paper would benefit from having additional baselines (for example [1-2] for active learning)”)*
> >
> > **A**: We appreciate your suggestions of the new baselines!
> >
> > Among the two references listed, Wei et al. (2015) focus on the batched active learning setting, and the goal is to remove redundancies within a mini-batch of queries---this is different from the sequential setting considered in our Task 2, which is equivalent to batch size = 1. Under this scenario (batch size = 1), their model reduces to uncertainty sampling, which we have already included as a baseline.
> >
> > As you suggested, we have added the BADGE algorithm as an additional baseline (Ash et al. 2020. Deep Batch Active Learning by Diverse, Uncertain Gradient Lower Bounds, ICLR 2020) for the active learning task on MNIST. We have observed that LeaSuRe still performs the best among all baselines. In particular, the performance of BADGE is better than *uncertainty sampling* and consistently worse than LeaSuRe; as the budget increases, the performance gap between BADGE and LeaSuRe becomes even larger.
> >
> > We will update Figure 2 in the revision to reflect the above results---does this address your concerns?

---

> > ### Comment · AnonReviewer3 · 2020-11-24
> > **Re: noisy set cover**
> >
> > For Q4, sorry for the misunderstanding. I was referring to the MSE results in Figure 4 of Appendix B, not Figure 1

---

> > > ### Author Response · Authors · 2020-11-25
> > > **Authors' response to the follow-up questions of AnonReviewer3**
> > >
> > > > For Q4, sorry for the misunderstanding. I was referring to the MSE results in Figure 4 of Appendix B, not Figure 1.
> > >
> > > Thanks for the clarification! In the Noisy Set Cover experiment, each label of the element added to the superset was perturbed with $N(0,1)$ noise. As a result, the variance of the total noise is linear in the number of sets. So, it is reasonable that the MSE error grows with the number of sets, as the policies cannot learn to predict random noise.
> > >
> > > Moreover, if you compare Figures 4a and 1a, you will notice that while stochastic MSE of LeaSuRe and the no-regularizer policy are similar, LeaSuRe outperforms the baselines in the number of elements added, which is what matters in practice. These two Figures confirm our intuition that when the problem is not exactly submodular, LeaSuRe will still generalize better than no regularizer by learning to ignore small deviations from submodularity.
> > > Finally, it is also expected that DSF has a lower MSE than LeaSuRe when the label noise is too large---Deep Submodular Functions are required to be submodular. When the stochasticity in the MSE becomes overwhelmingly large, that restrictive requirement becomes an advantage. However, when the MSE variance is not too large, the lack of expressiveness and the difficulty of optimization of DSF makes it lose its advantage compared to LeaSuRe. Since a lot of important real-life tasks are only slightly stochastic and are close to being submodular, we believe LeaSuRe offers a noticeable advantage compared to DSF.

---

### Author Response · Authors · 2020-11-25
**Summary of changes in rebuttal revision**

We thank the reviewers for their detailed comments and valuable suggestions.  We studied the reviews and discussions carefully and modified our paper accordingly.  Our revision followed the same list of actions proposed in our rebuttal response and further feedback from the reviewers.

Next we summarize the (key) changes included in [our revision](https://openreview.net/references/pdf?id=x9y8iA3J9), for each paper section.

**[1. Introduction]**

We have clarified our contribution in the revised introduction section. In particular, we highlighted that the protein engineering task in the experiment section is conducted as part of a collaboration with real protein biologists, and this method will be incorporated into the protein design workflow.

**[2. Related work]**

Following the suggestion by the reviewers, we added Batch Active Learning section to highlight its relationship to our problem.

**[5. Analysis]**

We updated Theorem 2 and its proof to incorporate feedback from AnonReviewer3. The proof is now more modular.

**[6. Experiments]**

We have included an additional baseline (BADGE, https://arxiv.org/abs/1906.03671) in Figure 2, as well as the discussion of different baselines in Baselines section and Appendix B. To address the earlier concern on our experimental setup, we now include the detailed description of the neural network architectures and the choice of hyperparameters for all experiments in Appendix B and Appendix C. We have additionally added clarifications in Section 6.3. to address the importance of our work as well as to describe the comparison of the protein engineering task to the baseline method in Liu et al. 2018.

---

### Decision · Program_Chairs · 2021-01-07
**Final Decision**

**Decision:**

Accept (Poster)

**Comment:**

This paper considers the problem of sequential decision making through the lens of submodular maximization. I read the paper myself and found the idea quite appealing and interesting. The authors also make a very effective rebuttal and brought a borderline paper into a clear accept.